# Study on the Bioactive Constituent and Mineral Elements of the Tibetan Medicine E’seguo from Different Regions of Ganzi Prefecture, China

**DOI:** 10.3390/molecules29174154

**Published:** 2024-09-01

**Authors:** Menglian Jiang, Heling Fan, Yixuan Chen, Yulin Zou, Xiaoyang Cai, Haohan Wang, Min Li

**Affiliations:** State Key Laboratory of Southwestern Chinese Medicine Resources, School of Pharmacy/School of Modern Chinese Medicine Industry, Chengdu University of Traditional Chinese Medicine, Chengdu 611137, China; 18382130605@163.com (M.J.); fhl19980902@126.com (H.F.); 13320872838@163.com (Y.C.); 18996255928@163.com (Y.Z.); caixycd@163.com (X.C.); wanghaohan@stu.cdutcm.edu.cn (H.W.)

**Keywords:** medicinal plants, E’seguo, bioactive constituent, mineral element

## Abstract

The Tibetan medicinal fruit E’seguo originates from two species, *Malus toringoides* (Rehd.) Hughes and *Malus transitoria* (Batal.) Schneid, both unique to the Hengduan Mountains. These species are predominantly found in high-altitude regions of Ganzi Prefecture, Sichuan Province, particularly in the Xianshui River and Yalong River basins. *Malus toringoides* (Rehd.) Hughes is far more abundant in both resource quantity and distribution compared to *Malus transitoria* (Batal.) Schneid. However, the nutritional and medicinal differences between the two remain unclear, which significantly impacts the development and utilization of E’seguo resources. This study aimed to measure the mineral content, nutritional components, and medicinal properties of E’seguo from 12 different regions of Ganzi Prefecture to explore the quality differences between these two species and across different regions. ICP-MS (Inductively Coupled Plasma Mass Spectrometry) was used to determine the mineral content, ultraviolet-visible spectrophotometry and potentiometric titration to analyze nutritional indicators, and HPLC (High-Performance Liquid Chromatography) to measure the medicinal components L-malic acid and 2-*O*-*β*-_D_-glucopyranosyl-_L_-ascorbic acid (AA-2*β*G). Results indicate that *Malus transitoria* (Batal.) Schneid contains higher levels of K, Ca, Zn, Mg, and Cu compared to *Malus toringoides* (Rehd.) Hughes, which has higher Fe and Mn content. *Malus toringoides* (Rehd.) Hughes from the Kangding and Litang regions showed the highest mineral content, with mineral elements primarily influencing polysaccharide levels, according to Mantel analysis. Nutritional and medicinal analyses revealed that *Malus toringoides* (Rehd.) Hughes outperformed *Malus transitoria* (Batal.) Schneid in all metrics except for the sugar-acid ratio. Given the mineral content and taste, *Malus transitoria* (Batal.) Schneid is better suited for consumption, while *Malus toringoides* (Rehd.) Hughes has superior medicinal properties, making it more appropriate for medicinal use. In the *Malus transitoria* (Batal.) Schneid regions, both Luhuo and Daofu are in the Xianshui River basin, with Daofu County producing the higher quality fruit. Among the nine *Malus toringoides* (Rehd.) Hughes regions, the M10 (Tuoba Township, Ganzi County) near the Yalong River had the highest overall score, followed by M7 (Yade Township, Luhuo County) and M6 (Keke, Xiala Tuo Town, Luhuo County), both of which are near the Xianshui River. In summary, *Malus transitoria* (Batal.) Schneid generally has higher mineral content, but *Malus toringoides* (Rehd.) Hughes has larger fruit and higher medicinal value, making the latter more suitable as a medicinal resource. At the same time, the medicinal quality of Xianshui River fruit was higher in the two watersheds of *Malus toringoides* (Rehd.) Hughes.

## 1. Introduction

E’seguo (its Tibetan medicine name is འོ་སེའི་འབྲས་བུ།) is the dried mature fruit of the *Malus toringoides* (Rehd.) Hughes or *Malus transitoria* (Batal.) Schneid of the Apple genus in the Rosaceae family, with a sweet and sour taste and neutral properties. It has the effects of clearing the lungs and resolving phlegm, nourishing the liver and improving eyesight, strengthening the stomach, and generating fluids. It is listed as a Tibetan medicine in the “Standards for Tibetan Medicinal Materials of Sichuan Province” (2020 edition). According to “Mirror of Tibetan Medicine Materia Medica” [1], E’seguo is described as follows: “The fruit is red like a rosehip, grows in high mountain valleys, harvested and sun-dried from August to October, round and red with wrinkled skin, containing seeds, sweet and sour in taste, and neutral in property.” In Ganzi Prefecture, E’seguo is also used as food and is a traditional medicinal and edible material. Ganzi Tibetan Autonomous Prefecture is located in the Hengduan Mountains of western Sichuan Province, China, and is one of the main distribution areas of Rosaceae Malus species in China [2]. *Malus toringoides* (Rehd.) Hughes is a member of the Rosaceae family, specifically the *Malus* genus within the subfamily Maloideae. The type specimen was collected in 1904 from western Sichuan, and Rehder described it as a variety of *Malus toringoides* (Rehd.) Hughes in 1915. In 1920, British botanist Hughes elevated it to species status [3]. According to Hua et al., *Malus toringoides* (Rehd.) Hughes is primarily distributed in the Sino-Himalayan Forest subregion, with its optimal distribution area in western Sichuan. In contrast, *Malus toringoides* (Rehd.) Hughes is mainly found in the Asian desert subregion, with its most suitable distribution area located at the junction of Sichuan, Qinghai, and Gansu provinces [4].

In a previous study by our research group on its phenolic acid components, 2-*O*-*β*-_D_-glucopyranosyl-_L_-ascorbic acid (AA-2*β*G) [5] was discovered in E’seguo for the first time, and its lipid-lowering pharmacological effects were studied [6]. The results showed that E’seguo could significantly reduce the levels of Total Cholesterol (TC) and Triglycerides (TG) in lipid-laden cells and improve their morphology. It also significantly lowered the liver index and the levels of TC, TG, Low-Density Lipoprotein Cholesterol (LDL-C), Alanine Aminotransferase (ALT), and Aspartate Aminotransferase (AST) in hyperlipidemic mice while increasing High-Density Lipoprotein Cholesterol (HDL-C) levels. However, current research on its nutritional value and bioactive components is limited. Therefore, this study combines mineral elements, nutritional components, and medicinal components to analyze the quality differences of E’seguo from two species and 12 regions in Ganzi Prefecture, comprehensively evaluating the medicinal and edible value of E’seguo.

The Apple genus in the Rosaceae family includes many fruit-bearing genera, including apples, pears, plums, and peaches. Apples are especially rich in various nutrients, such as polysaccharides, polyphenols, pectin, vitamins, and abundant phenolic antioxidants [7,8]. Total polysaccharides are high molecular compounds formed by many monosaccharides through glycosidic bonds. Apple pomace contains soluble cell wall substances with high colloidal value, mainly composed of galacturonic acid, arabinose, galactose, and a small number of other sugars [9]. Additionally, apple skin polysaccharides have been identified as heteropolysaccharides rich in arabinose, galactose, and galacturonic acid, with antioxidant activity and hepatoprotective effects [10]. Extracted apple polysaccharides induce apoptosis in colorectal cancer cells through the NF-ωB pathway, showing potential anti-cancer effects, making them a promising drug for cancer prevention and treatment [11]. Apple peels contain many phenolic compounds, such as chlorogenic acid, catechin, and epicatechin [12], with antioxidant properties and various health benefits [13]. Additionally, apples contain various organic acids such as tartaric acid and malic acid [14,15], with significant nutritional value.

Mineral elements play a crucial role in the growth, development, and quality of Rosaceae fruits. Studies have shown that mineral nutrition, including potassium, calcium, phosphorus, iron, magnesium, and other elements, directly affects fruit quality and post-harvest storage [16]. The composition and accumulation of mineral components in fruits vary significantly at different stages. For example, a study on the Australian native fruit broad-leaved geebung highlighted its composition of minerals and trace elements, with calcium, potassium, iron, zinc, and manganese identified as the main elements, showing significant differences at different growth stages [17]. Research on *P. ussuriensis* fruit showed higher contents of minerals such as potassium, calcium, and magnesium compared to other pear varieties, underscoring the importance of mineral analysis for selecting and breeding fruit varieties [18]. Mineral elements such as potassium, calcium, iron, manganese, and zinc influence the metabolism, photosynthesis, and defense mechanisms of fruit trees [19]. Foliar application of minerals such as calcium and magnesium has been found to improve fruit set, retention, yield attributes, and quality while reducing fruit drop in fruit crops [19]. Additionally, the content of iron, manganese, zinc, and copper in fruits contributes significantly to their phytochemical and antioxidant properties, and overall nutritional value [16]. These studies collectively emphasize the importance of mineral elements in fruits for human health and well-being.

## 2. Results

### 2.1. Mineral Elements Analysis of E’seguo

The standard curves for L-malic acid and AA-2*β*G were plotted using the concentration of the reference substance as the *x*-axis and the peak area as the *y*-axis. The results are shown in Table 1. The results indicate that L-malic acid exhibited good linearity in the range of 0.626 mg/mL to 1.565 mg/mL, and AA-2*β*G exhibited good linearity in the range of 0.031 mg/mL to 0.76 mg/mL.

The results of mineral element determination in E’seguo are shown in Figure 1. The content of K ranged from 3.89 to 5.44 g/kg, with an average of 4.90 g/kg, a median of 4.92 g/kg, and a coefficient of variation of 5.82%. The Ca content ranged from 0.89 to 2.32 g/kg, with an average of 1.55 g/kg, a median of 1.51 g/kg, and a coefficient of variation of 18.43%. Zn content ranged from 31.11 to 102.15 μg/kg, with an average of 52.80 μg/kg, a median of 50.93 μg/kg, and a coefficient of variation of 32.95%. Mg content ranged from 4.27 to 10.23 μg/kg, with an average of 6.50 μg/kg, a median of 6.20 μg/kg, and a coefficient of variation of 20.22%. Cu content ranged from 4.22 to 13.52 μg/kg, with an average of 8.88 μg/kg, a median of 8.88 μg/kg, and a coefficient of variation of 27.03%. Fe content ranged from 0.53 to 2.03 μg/kg, with an average of 0.98 μg/kg, a median of 0.94 μg/kg, and a coefficient of variation of 31.55%. Se content ranged from 0.63 to 2.16 μg/kg, with an average of 1.22 μg/kg, a median of 1.13 μg/kg, and a coefficient of variation of 31.88%. Mn content ranged from 0.24 to 0.73 μg/kg, with an average of 0.53 μg/kg, a median of 0.54 μg/kg, and a coefficient of variation of 22.05%.

Overall, the fruits of *Malus toringoides* (Rehd.) Hughes have higher contents of K, Cu, Fe, and Mn compared to *Malus transitoria* (Batal.) Schneid, while *Malus transitoria* (Batal.) Schneid fruits have higher contents of Ca, Zn, Mg, and Se than *Malus toringoides* (Rehd.) Hughes.

**Figure 1 molecules-29-04154-f001:**
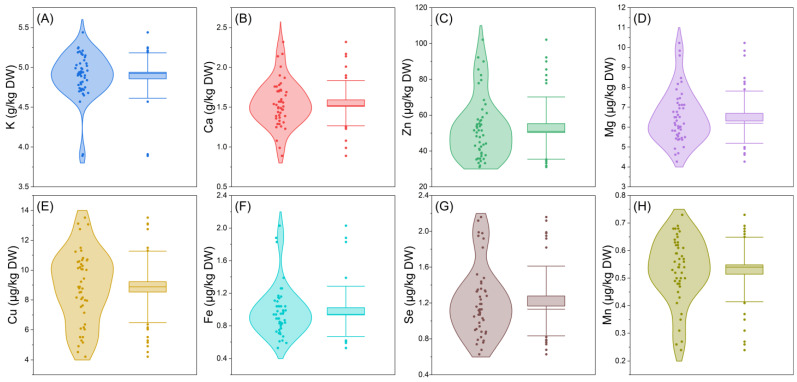
Violin plots of the mineral elements of E’seguo. In these plots, the whiskers of the box plot represent the standard deviation (SD), the top and bottom of the box represent the standard error (SE), and the middle horizontal line indicates the median. This figure covers the following measured mineral elements, (**A**) is K (potassium), (**B**) is Ga (calcium), (**C**) is Zn (zinc), (**D**) is Mg (magnesium), (**E**) is Cu (copper), (**F**) is Fe (iron), (**G**) is Se (selenium), and (**H**) is Mn (manganese).

In E’seguo, the highest K content was found in M9 (5.07 g/kg) and the lowest in M5 (3.72 g/kg). The highest Ca content was in M4 (2.16 g/kg) and the lowest in M9 (1.19 g/kg). The highest Zn content was in M4 (91.71 μg/kg) and the lowest in M5 (34.46 μg/kg). The highest Mg content was in M4 (8.57 μg/kg) and the lowest in M10 (5.20 μg/kg). The highest Cu content was in M3 (11.24 μg/kg) and the lowest in M5 (4.65 μg/kg). The highest Fe content was in M9 (1.78 μg/kg) and the lowest in M4 (0.66 μg/kg). The highest Se content was in M4 (2.01 μg/kg) and the lowest in M5 (0.69 μg/kg). The highest Mn content was in M9 (0.68 μg/kg) and the lowest in M4 (0.27 μg/kg).

In Figure 2, among the mineral elements, Mg, Ca, Se, and Zn were clustered together, while Fe, K, Mn, and Cu were grouped together. Systematic cluster analysis results showed that the E’seguo samples from the 12 production areas could be divided into three categories. The first major category consisted of M4, which was Malus transitoria from Daofu County. These fruits were yellowish-white, smaller, and sour, with the highest Ca, Zn, Mg, and Se content and the lowest Fe and Mn content among the 12 areas. The second category included M9 and M12, which were Malus toringoides from Kangding City and Litang County, respectively. These fruits were obovate, reddish-yellow, larger, and from regions with similar latitudes, which were lower compared to other areas. The M9 region (Kangding City) had the highest K, Fe, and Mn content, while M12 (Litang County) had relatively high Zn and Se content. The remaining areas formed the third category, encompassing nine production areas from Luhuo County, Daofu County, Ganzi County, and Xinlong County.

In Figure 3, Principal Component Analysis (PCA) revealed that PC1 accounted for 47.72% and PC2 accounted for 25.10% of the variance, cumulatively explaining 72.82% of the information content regarding the mineral elements in E’seguo. Thus, these two principal components can be used to evaluate the quality of E’seguo from the 12 production areas. In the loading plot, Zn and Se were the most significant for explaining the M4 and M12 production areas. M1, M2, M3, M5, M6, M7, and M10 clustered together; M9 formed a separate cluster; M4 and M12 clustered together.

In the correlation analysis of Figure 4, K showed a significant negative correlation with Mg (−0.48) and a significant positive correlation with Mn (0.29). Ga was significantly positively correlated with Zn and Mg (0.42, 0.51) and significantly negatively correlated with Fe and Mn (−0.56, −0.37). Zn was significantly positively correlated with Se (0.72) and significantly negatively correlated with Mn. Mg showed significant negative correlations with Cu and Mn (−0.33, −0.46). Cu was significantly negatively correlated with Fe (−0.59). Fe had a significant positive correlation with Mn (0.43). Se was significantly negatively correlated with Mn (−0.55).

### 2.2. Analysis of Quality Indicators and Medicinal Properties of E’seguo

The quality indicators of E’seguo include total polysaccharides, total acid, sugar-acid ratio, and total polyphenol, L-malic acid, AA-2*β*G. The results of determination of the quality indicators of E’seguo are shown in Figure 5. The total polysaccharide content ranged from 16.35% to 38.50%, with an average of 24.63%, a median of 24.65%, and a coefficient of variation of 17.37%. The total acid content ranged from 0.81% to 3.34%, with an average of 2.12%, a median of 2.15%, and a coefficient of variation of 32.50%. The sugar-acid ratio ranged from 7.21 to 31.69, with an average of 13.44, a median of 11.05, and a coefficient of variation of 49.50%. The total polyphenol content ranged from 1.27% to 3.81%, with an average of 2.30%, a median of 2.19%, and a coefficient of variation of 23.14%. The L-malic acid content ranged from 0.65% to 2.13%, with an average of 1.29%, a median of 1.37%, and a coefficient of variation of 28.06%. The AA-2*β*G content ranged from 1.56% to 4.55%, with an average of 3.15%, a median of 3.20%, and a coefficient of variation of 23.03%.

In E’suo, the highest total polysaccharide content was found in M6 (32.44%) and the lowest in M4 (17.14%). The highest total acid content was in M10 (3.05%) and the lowest in M8 (0.88%). The highest sugar-acid ratio was in M8 (28.88) and the lowest in M7 (8.08). The highest total polyphenol content was in M10 (3.34%) and the lowest in M5 (1.45%). The highest L-malic acid content was in M10 (4.08%) and the lowest in M5 (1.56%). The highest AA-2βG content was in M1 (1.71%) and the lowest in M8 (0.78%).

Systematic cluster analysis results (Figure 6) showed that the E’seguo samples from the 12 production areas could be divided into three major categories: The first category included M10, M7, M3, and M12, all *Malus toringoides* (Rehd.) Hughes from Ganzi County, Yade Township in Luhuo County, Mazi Township in Daofu County, and Litang County. These fruits were completely red or reddish-yellow, with smooth obovate skins and larger sizes. This category had high contents of total acid, total polyphenols, AA-2*β*G, and L-malic acid, making the fruits taste sour and astringent. The second category included M6, M9, M11, M1, and M2, all *Malus toringoides* (Rehd.) Hughes from Xialatuo in Luhuo County, Kangding City, Xinlong County, Kose Township in Daofu County, and Warizhen in Daofu County. These fruits were oval or obovate with smooth skins and completely red or reddish-yellow in color. This category had high contents of total polysaccharides and sugar-acid ratios, resulting in a sweet and sour taste. The third category included M4, M5, and M8, consisting of *Malus transitoria* (Batal.) Schneid from Daofu County, *Malus toringoides* (Rehd.) Hughes from Xialatuo in Luhuo County, and *Malus transitoria* (Batal.) Schneid from Luhuo County. Although M5 was *Malus toringoides* (Rehd.) Hughes, its fruit characteristics and size were like those of *Malus transitoria* (Batal.) Schneid. These fruits were mostly yellowish-white with some red blushes and red spots, cylindrical or apple-shaped, and smaller in size. This category had lower content in all components and thus lower nutritional value.

In the correlation analysis of Figure 7, total polysaccharides and the sugar-acid ratio showed a significant positive correlation (0.32, 0.32). Total acid had a significant negative correlation with the sugar-acid ratio (−0.86) and significant positive correlations with total polyphenols, L-malic acid, and AA-2*β*G (0.70, 0.31, 0.83). The sugar-acid ratio was significantly negatively correlated with total polyphenols, L-malic acid, and AA-2*β*G (−0.48, −0.30, −0.71). Total polyphenols showed significant positive correlations with L-malic acid and AA-2*β*G (0.59, 0.47). In the Mantel analysis, mineral elements significantly influenced total polysaccharides.

Principal component analysis (PCA) was performed on the quality indicators and medicinal properties of E’seguo. The results are shown in Figure 8 and Table 2. The first principal component (PC1) accounted for 58.88%, the second (PC2) for 25.63%, and the third (PC3) for 9.15% of the variance. Together, these three principal components explained 93.66% of the variance, representing most of the information content in the quality and medicinal indicators of E’seguo. Therefore, these three principal components can be used to evaluate the quality and medicinal properties of E’seguo from the 12 production areas.

PC1 primarily integrates the information from total acid, total polyphenols, AA-2*β*G, and malic acid. These indicators show a positive distribution on the first principal component, meaning that the higher the PC1 value, the higher the values of these indicators. Hence, PC1 mainly reflects the phenolic acid content in E’seguo, with higher values indicating a more sour and astringent taste, making it more suitable for medicinal use rather than as food. PC2 primarily integrates the information from total polysaccharides, malic acid, total polyphenols, and the sugar-acid ratio, mainly reflecting the sweet and sour taste of E’seguo. A higher PC2 value indicates a sweeter taste, making this type of E’seguo more suitable for consumption. PC3 mainly reflects the total polysaccharides and AA-2*β*G content.

Based on the principal component analysis model, the eigenvectors for each indicator were calculated and used as weights to derive the scoring formulas for the three principal components (H_1_, H_2_, H_3_). SPSS software was used to standardize the quality data of the fruit and, inputting it into these scoring formulas, we obtained the scores of E’seguo from different production areas on six principal components. The weights for these principal components were based on their respective variance contribution rates, and the comprehensive quality score (H_0_) of the fruit was calculated accordingly.

H_1_ = 0.022Z_1_ + 0.512Z_2_ − 0.472Z_3_ + 0.460Z_4_ + 0.314Z_5_ + 0.451Z_6_

H_2_ = 0.730Z_1_ − 0.100Z_2_ + 0.269Z_3_ + 0.314Z_4_ + 0.456Z_5_ − 0.279Z_6_

H_3_ = 0.521Z_1_ + 0.233Z_2_ + 0.131Z_3_ + 0.094Z_4_−0.756Z_5_ + 0.278Z_6_

H_0_ = 0.63H_1_ + 0.27H_2_ + 0.10H_3_

Table 3 shows that the comprehensive scores for each production area are ranked as follows: M10 > M7 > M6 > M12 > M3 > M9 > M1 > M11 > M2 > M4 > M5 > M8. Overall, the quality of E’seguo from *Malus toringoides* (Rehd.) Hughes is superior to that of *Malus transitoria* (Batal.) Schneid (M4, M5). Among the 12 production areas, the E’seguo from Ganzi County has the highest quality. The E’seguo from M5 (Renda Township, Luhuo County) has relatively poor quality. The analysis suggests that the fruit in this area resembles that of *Malus transitoria* (Batal.) Schneid, with fresh fruit being yellowish-white with a few red tinges and spots, and smaller in size. In contrast, E’seguo from other areas are mostly completely red or red dish-yellow and larger in size.

## 3. Materials and Methods

### 3.1. Plant Material

The primary species distributed in Ganzi Prefecture is *Malus toringoides* (Rehd.) Hughes, while *Malus transitoria* (Batal.) Schneid is sparsely distributed. *Malus toringoides* (Rehd.) Hughes is mainly concentrated in the Xianshui River basin, which eventually merges into the Yalong River system of the Yangtze River Basin (Figure 9), covering areas such as Luhuo, Daofu, Ganzi, Xinlong, Yajiang, Kangding, and Litang. On the other hand, *Malus transitoria* (Batal.) Schneid is primarily found in the Xianshui River basin, specifically in the Luhuo and Daofu counties. The fruit of E’seguo is harvested around October each year. During October, the average precipitation rate is about 0.14 mm/h, gradually decreasing thereafter, with rainfall peaking between June and August. The average temperature in October is relatively low [20].

Samples of mature E’seguo fruits from *Malus toringoides* (Rehd.) Hughes or *Malus transitoria* (Batal.) Schneid were collected in October 2023 from twelve production areas across six counties in Ganzi Prefecture, Sichuan Province, China. Four trees were randomly selected from each production area, resulting in four replicates per area. The collected samples were dried at 55 °C until they reached a constant weight, then ground into powder and passed through a 250 μm sieve for analysis. Detailed sampling information and varieties from each production area are listed in Table 4. The different origins of E’seguo are shown in Figure 10. From the appearance traits, *Malus toringoides* (Rehd.) Hughes fresh fruit has smooth, wax-like, red or reddish-yellow skin. The fruit is obovate-shaped or oblong oval, 1–1.3 cm in diameter. *Malus transitoria* (Batal.) Schneid fresh fruit has Pome yellowish-red, subglobose or oblong-ellipsoid fruit, 6–8 mm in diam. The fruits are nearly spherical or cylindrical and smaller in size.

### 3.2. Determination of Mineral Elements, Polysaccharides, Total Acids, Sugar-Acid Ratio, and Polyphenols

The content of mineral elements (K, Ca, Mg, Fe, Mn, Cu, Zn, Se) in E’seguo was determined using inductively coupled plasma mass spectrometry (ICP-MS) [21]. The total polysaccharides were extracted using an improved alcohol precipitation method [22]. Dried E’seguo powder was mixed with 95% ethanol at a material-to-liquid ratio of 1:100, refluxed for 45 min, and filtered. The residue and filter paper were then mixed with water at a material-to-liquid ratio of 1:150, ultrasonicated for 1 h, and filtered to obtain the extract. The total polysaccharide content was measured using the sulfuric acid-anthrone colorimetric method. Total acid content was determined using potentiometric titration with a pH meter [23]. The sugar-to-acid ratio was calculated using the formula total polysaccharides/total acid. For total polyphenol determination, 1.0 g of fine E’seguo powder was mixed with 50% ethanol at a material-to-liquid ratio of 1:20, refluxed at 80 °C for 75 min, and filtered. A 1 mL aliquot of the filtrate was diluted to 10 mL with water, and then 1 mL of this solution was further diluted to 10 mL with water. This diluted solution was mixed with 3 mL of 5% sodium carbonate solution and 0.5 mL of Folin-Ciocalteu reagent, then diluted to 10 mL with water and mixed. The absorbance was measured at 740 nm after incubating at 75 °C for 50 min. Ultra-pure water treated similarly was used as a blank control. The total polyphenol content was calculated based on the absorbance readings. The methodological investigation of total polysaccharides and total polyphenols is described in the Appendix A.

### 3.3. Determination of L-malic Acid and AA-2βG

Standards for L-malic acid (reference substance number Wkq21021904) and AA-2*β*G (reference substance number Wkq20042702) were purchased from Sichuan Weikeqi Biological Technology Co., Ltd. (Chengdu, China) High-performance liquid chromatography (HPLC)-grade methanol, acetonitrile, and phosphoric acid were obtained from Sichuan Cologne Chemical Co., Ltd (Chengdu, China). HPLC was used to determine the content of L-malic acid and AA-2*β*G in the samples.

For AA-2*β*G extraction, 0.5 g of E’seguo powder was placed in a conical flask with a stopper, and 25 mL of water was added. The mixture was ultrasonicated (500 W, 40 k Hz) for 45 min, shaken well, and filtered to obtain the filtrate. The chromatographic conditions for AA-2*β*G included using a C_18_-bonded silica gel column (250 mm length, 4.6 mm inner diameter, 5 µm particle size). The mobile phase consisted of acetonitrile (A) and 0.1% phosphoric acid solution (B), with a gradient elution as follows: 0–10 min, 2% acetonitrile; 10–11 min, 2%→5% acetonitrile; 11–17 min, 5% acetonitrile; 17–20 min, 5%→10% acetonitrile; 20–22 min, 10%→2% acetonitrile; 22–25 min, 2% acetonitrile. Detection was carried out at 235 nm, with a flow rate of 0.5 mL/min and a column temperature of 30 °C. The retention time for AA-2*β*G was 8.1 min.

For L-malic acid extraction, 0.3 g of E’seguo powder was accurately weighed and placed in a conical flask with a stopper. Exactly 10 mL of distilled water was added, and the weight was recorded. The mixture was ultrasonicated (500 W, 40 k Hz) for 20 min and cooled, and the weight was recorded again. Distilled water was added to make up for any lost weight, and the mixture was shaken well, and filtered to obtain the filtrate. The chromatographic conditions for L-malic acid were similar, using a C_18_-bonded silica gel column (250 mm length, 4.6 mm inner diameter, 5 µm particle size). The mobile phase consisted of acetonitrile (A) and 0.1% phosphoric acid solution (B), with a gradient elution as follows: 0-3 min, 1%→2% acetonitrile; 3-10 min, 2% acetonitrile; 10-11 min, 2%→4% acetonitrile; 11-13 min, 4%→15% acetonitrile; 13-16 min, 15%→31% acetonitrile; 16-20 min, 31%→1% acetonitrile. Detection was carried out at 235 nm, with a flow rate of 0.5 mL/min and a column temperature of 30 °C. The retention time for L-malic acid was 12.4 min. The methodological investigations of L-malic acid and AA-2*β*G are described in the Appendix A.

### 3.4. Statistical Analysis

All data were processed using Excel. Violin plots were created using Origin 2024b, with the curve type set to Kernel Smooth, and the whiskers of the box plot representing the standard deviation (SD) and the top and bottom of the box representing the standard error (SE). Principal component analysis (PCA) was also performed, with all data analysis completed in Origin. Clustering was carried out using the R package ‘*Pheatmap*’ with the clustering method set to complete and the distance measured using Euclidean. Correlation analysis was performed using the R package ‘*Hmisc*’ with the Pearson method. Mantel test analysis was used to test the correlation between two matrices and is often used to analyze the correlation of different distance matrices. The significance of the correlation was determined by evaluating the Z-statistic of the actual data by permuting the matrix and calculating the correlation coefficient. Mantel test analysis was conducted using the R package ’*Vegan*’ utilizing the Mantel method with Bray-Curtis dissimilarity and Euclidean distance. The relevant plots in R were created using the online platform www.chiplot.online (accessed on 10 July 2024). One-way ANOVA was performed using SPSS 26.0, with Duncan’s new multiple range test employed for multiple comparisons.

## 4. Discussion

*Malus toringoides* (Rehd.) Hughes and *Malus transitoria* (Batal.) Schneid are both plants of the Rosaceae family and Malus genus. Studies have shown that *Malus transitoria* (Batal.) Schneid is one of the progenitors of *Malus toringoides* (Rehd.) Hughes [24], which explains their similar morphological characteristics and overlapping distribution areas. The high-altitude regions of western Sichuan Province are the main distribution areas for both species, and they are endemic species to the Hengduan Mountains region in China. Local Tibetan people often dry their young leaves to make tea or ferment them through a specific process to make brick tea for drinking. Their fruits are often used to make fruit wine, which has a unique flavor and is popular among the local Tibetans [25]. Rosaceae plants are a large family of approximately 3000 species, including many economically important fruits such as apples, strawberries, peaches, plums, and apricots [26], as well as various crabapples like Hubei crabapple, Siberian crabapple, and Lijiang crabapple. These crabapples are not only beautiful ornamental plants but also have medicinal uses [27]. Apples, being of the same genus as E’seguo, are rich in polyphenols, polysaccharides, proteins, amino acids, and volatile substances, and have antioxidant, hypoglycemic, hypolipidemic, anticancer, and anti-inflammatory activities [28]. Researchers have used ICP-AES (Inductively coupled plasma atomic emission spectrometry) and AFS (Atomic fluorescence spectroscopy) to determine the mineral elements of 565 apples from the southwestern plateau of China. Their K, Ca, and Mg contents were relatively high, with averages of 1001.94, 45.00, and 49.46 mg/kg, respectively; Fe, Mn, Cu, and Zn were moderate, at 1.70, 0.34, 0.27, and 0.24 mg/kg, respectively; Se content was relatively low at 0.0039 mg/kg [29]. The average contents of K, Cu, Fe, Mn in the ten regions of *Malus toringoides* (Rehd.) Hughes were 4.84 mg/kg, 8.81 μg/kg, 1.01 μg/kg, and 0.55 μg/kg, respectively, while for *Malus transitoria* (Batal.) Schneid, the contents were 4.79 g/kg, 8.69 μg/kg, 0.71 μg/kg, and 0.41 μg/kg, respectively. The average contents of Ca, Zn, Mg, Se in *Malus toringoides* (Rehd.) Hughes were 1.45 g/kg, 49.59 μg/kg, 6.15 μg/kg, and 1.12 μg/kg, respectively, while in *Malus transitoria* (Batal.) Schneid, they were 1.88 g/kg, 68.68 μg/kg, 7.52 μg/kg, and 1.63 μg/kg, respectively [30]. Overall, the fruit of *Malus toringoides* (Rehd.) Hughes had higher contents of K, Cu, Fe, and Mn compared to *Malus transitoria* (Batal.) Schneid, while *Malus transitoria* (Batal.) Schneid had higher contents of Ca, Zn, Mg, and Se than *Malus toringoides* (Rehd.) Hughes. Among these, the contents of K and Ca were higher than other elements. As fruits of the Malus genus, *Malus toringoides* (Rehd.) Hughes, like other apples, are high-potassium fruits. Potassium can balance sodium ions in the body, maintain blood pressure stability, and reduce the risk of cardiovascular diseases [31]. Thus, E’seguo can also be a good source of potassium supplementation. A systematic cluster analysis of E’seguo from 12 regions showed that they could be divided into three major categories: M4 (flowering crabapple from Daofu County) formed the first category; M9 and M12 (*Malus toringoides* (Rehd.) Hughes from Kangding City and Litang County) clustered into the second category; the rest, including Luohuo County, Daofu County, Ganzi County, and Xinlong County, formed the third category, with nine regions in total. There were significant differences in the mineral element contents among the three categories.

As it is a traditional medicinal and edible plant in the Tibetan region, it is crucial to elucidate not only the nutritional value of E’seguo but also to explore its medicinal potential. Rosaceae plants mainly contain polysaccharides, polyphenols, phenolic acids, and flavonoids as their nutritional components. Polysaccharides are key natural macromolecules in the fruits of Rosaceae plants, composed of large numbers of monosaccharides linked by glycosidic bonds. They play significant roles in the development and metabolism of plants and are important indicators of fruit quality and nutritional value. Common fruits in the same genus as E’seguo, such as apples, contain primarily glucose, fructose, sucrose, and other free sugars, with a total sugar content of 11.5–16.00% FW [32]. In comparison, the total polysaccharide content of *Malus toringoides* (Rehd.) Hughes (24.92%) is higher than that of *Malus transitoria* (Batal.) Schneid (21.18%), with the highest total polysaccharide content found in the M6 region (Xiala Town, Luohuo County) at 32.44%. Compared to cultivated fruits used for consumption, such as apples and crabapples, E’seguo fruits are smaller with limited edible parts and lower total polysaccharide content, but higher than other crabapple species in the same genus (average content 0.16–0.48% FW) [33]. Total acid content and the sugar-acid ratio are crucial indicators of fruit flavor. The balance between total polysaccharides and total acids affects the taste of fruits and serves as a valuable diagnostic tool for monitoring fruit quality during storage. Generally, fruits with a balanced sweet and sour taste are preferred, with a sugar-acid ratio of about 20–60. A sugar-acid ratio below 20 indicates a bland or sour taste, while above 60, the sweetness is enhanced. Wild apples typically have significantly higher total acid content (0.8–22.7 mg/g) compared to cultivated apples (0.5–18.9 mg/g), with an average difference of more than 2.2 times. The total sugar content of cultivated apples (93.1 mg/g) is slightly higher than that of wild apples (89.9 mg/g), but the difference is not significant [34]. This indicates that within the same species, cultivated varieties have higher total polysaccharide content and lower total acid content, making the sugar-acid ratio more suitable for taste. As a wild plant in the Tibetan region of Ganzi Prefecture, the total polysaccharide content of *Malus toringoides* (Rehd.) Hughes (24.92%) is higher than that of *Malus transitoria* (Batal.) Schneid (21.18%). However, the total acid content of *Malus toringoides* (Rehd.) Hughes (2.24%) is also higher than that of *Malus transitoria* (Batal.) Schneid (1.41%), resulting in a lower sugar-acid ratio (11.16) compared to *Malus transitoria* (Batal.) Schneid (15.02). Therefore, in terms of taste, *Malus transitoria* (Batal.) Schneid is more suitable for consumption. Future breeding of E’seguo for edible purposes should focus on selecting *Malus transitoria* (Batal.) Schneid and reducing its total acid content to improve the sugar-acid ratio, thereby producing cultivars with a balanced sweet and sour taste.

Studies on polyphenols in Rosaceae fruits like apples, peaches, pears, and plums have shown that plums have the highest total phenolic content, making them a rich source of natural antioxidants [35]. Apples, particularly the peels, contain various polyphenolic compounds such as flavanols, phenolic acids, dihydrochalcones, flavonols, and anthocyanins, which contribute to their antioxidant capacity. Research indicates that apple polyphenols (AP) can prevent oxidative stress-induced damage by enhancing cell viability, increasing antioxidant enzyme activity, reducing oxidation products, inhibiting DNA fragmentation, and decreasing the expression of apoptosis-related proteins [36]. In E’seguo, the total polyphenol content in *Malus toringoides* (Rehd.) Hughes (2.41%) is higher than in *Malus transitoria* (Batal.) Schneid (1.62%). Polyphenols contribute to astringency and higher antioxidant capacity. Thus, considering all nutritional indicators measured, *Malus transitoria* (Batal.) Schneid is more suitable for consumption compared to *Malus toringoides* (Rehd.) Hughes. E’seguo, when used in Tibetan medicine, exhibits pharmacological effects such as lowering blood lipids, reducing blood sugar, and providing antioxidant properties [6]. The active compounds responsible for these effects include organic acids like malic acid, citric acid, and AA-2*β*G. Malic acid can reduce lipid peroxidation in the liver and heart of aged rats, increase antioxidant enzyme activity, and help control hyperuricemia [37]. *Malus toringoides* (Rehd.) Hughes has a higher content of L-malic acid (3.12%) compared to *Malus transitoria* (Batal.) Schneid (3.10%), with the highest concentration found in Kangding City (M9). AA-2*β*G, a natural vitamin C analog, was first discovered in E’seguo and possesses good antioxidant activity, lipid-lowering effects, and hepatoprotective properties [38]. The AA-2*β*G content in wolfberry ranges from 0.75% to 1.55% [39], and in E’seguo, it is 1.37% for *Malus toringoides* (Rehd.) Hughes and 0.80% for *Malus transitoria* (Batal.) Schneid, making E’seguo a potential source for AA-2*β*G extraction. Considering both medicinal indicators, *Malus toringoides* (Rehd.) Hughes is superior to *Malus transitoria* (Batal.) Schneid and is more suitable for medicinal use. Through principal component analysis, the nutritional and medicinal indicators of E’seguo from 12 regions were comprehensively evaluated. The two *Malus transitoria* (Batal.) Schneid production regions are both located in the Xianshui River basin, but Daofu County’s *Malus transitoria* (Batal.) Schneid scored higher overall. Among the nine *Malus toringoides* (Rehd.) Hughes production regions, which include the Xianshui River and Yalong River basins as well as some hillside areas, E’seguo from the Yalong River basin in Ganzi County achieved the highest comprehensive score. Following this, the Yade Township and Xiala Tuo Town regions in Luhuo County, both located in the Xianshui River basin, also ranked highly. Therefore, *Malus toringoides* (Rehd.) Hughes grown in regions near rivers tends to exhibit superior quality.

## 5. Conclusions

In this study, we conducted an analysis of the mineral elements, nutritional components, and medicinal properties of E’seguo (from *Malus toringoides* (Rehd.) and *Malus transitoria* (Batal.) Schneid) collected from 12 regions in Ganzi Prefecture. In terms of mineral elements, *Malus transitoria* (Batal.) Schneid had higher levels of K, Ca, Zn, Mg, and Cu, while *Malus toringoides* (Rehd.) Hughes showed higher concentrations of Fe and Mn. For both nutritional and medicinal components, *Malus toringoides* (Rehd.) Hughes outperformed *Malus transitoria* (Batal.) Schneid, although the two species exhibited minimal differences in total polysaccharide content. However, the sugar-acid ratio in *Malus transitoria* (Batal.) Schneid was higher, indicating a sweeter and more palatable taste, making it more suitable for consumption. Conversely, *Malus toringoides* (Rehd.) Hughes is better suited for medicinal use. The primary *Malus transitoria* (Batal.) Schneid production areas are in the Xianshui River basin, specifically Luhuo County and Daofu County. Principal component analysis revealed that *Malus transitoria* (Batal.) Schneid from Daofu County scored higher than that from Luhuo County. Among the nine *Malus toringoides* (Rehd.) Hughes production regions, the highest overall score was achieved by E’seguo from Ganzi County, followed by the Yade Township and Xiala Tuo Town regions in Luhuo County, all of which are located near rivers. At the same time, the medicinal quality of Xianshui River was higher in the two watersheds of *Malus toringoides* (Rehd.) Hughes.

## Figures and Tables

**Figure 2 molecules-29-04154-f002:**
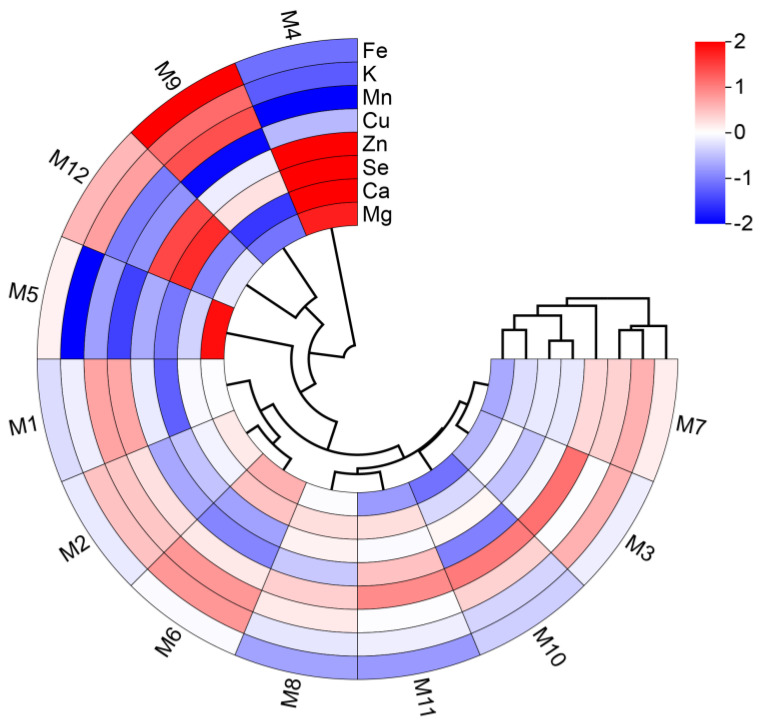
Cluster dendrogram of the mineral elements of E’seguo.

**Figure 3 molecules-29-04154-f003:**
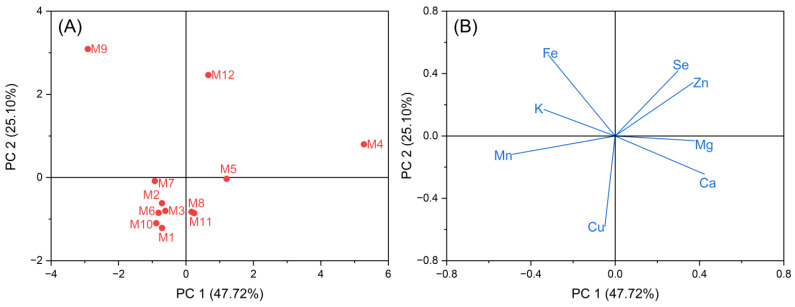
Principal Component Analysis (PCA) of the mineral elements of E’seguo. (**A**) is the score plot, and (**B**) is the loading plot.

**Figure 4 molecules-29-04154-f004:**
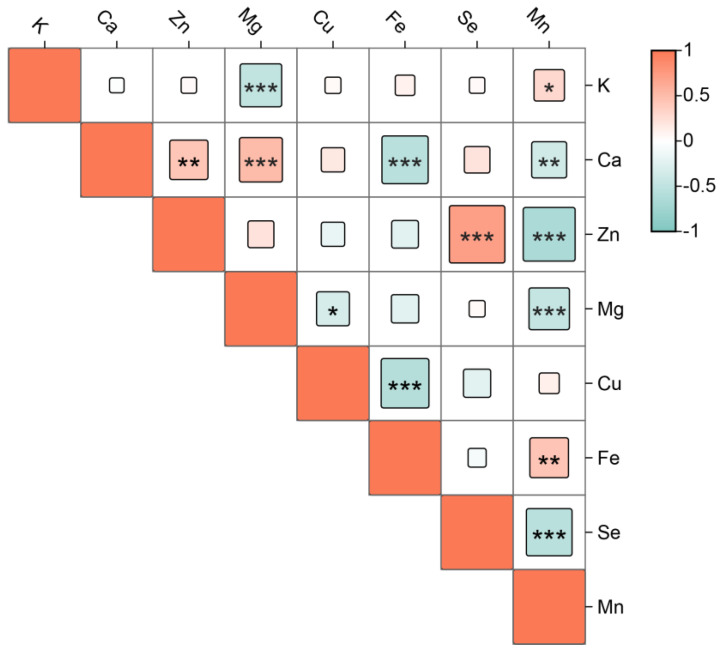
Correlation analysis of mineral elements. * Indicates *p* < 0.05, ** indicates *p* < 0.01, and *** indicates *p* < 0.001.

**Figure 5 molecules-29-04154-f005:**
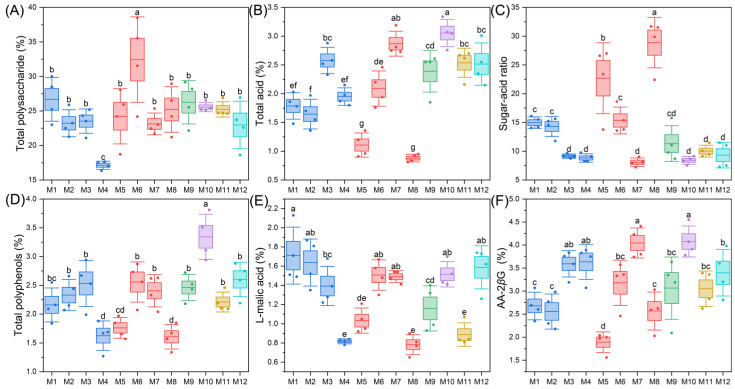
Box plots of the quality indicators and medicinal properties of E’seguo. In these plots, the whiskers of the box plot represent the standard deviation (SD), the top and bottom of the box represent the standard error (SE), and the middle horizontal line indicates the mean. This figure covers the following measurement indicators, (**A**) is total polysaccharide, (**B**) is total acid, (**C**) is sugar acid ratio, (**D**) is total polyphenols, (**E**) is L-malic acid, (**F**) is AA-2βG. Different lowercase letters indicate differences at the *p* < 0.05 level.

**Figure 6 molecules-29-04154-f006:**
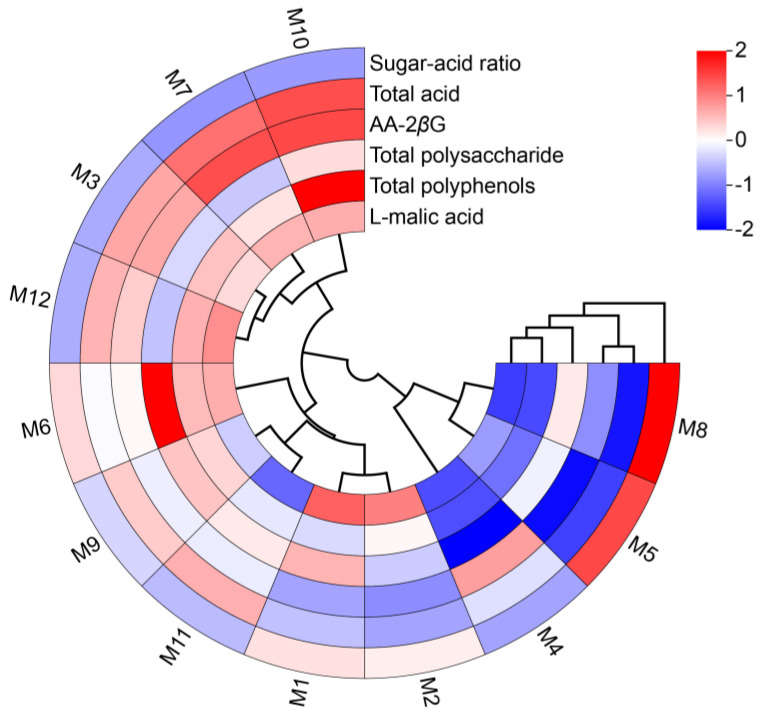
Cluster dendrogram of the quality indicators and medicinal properties of E’seguo.

**Figure 7 molecules-29-04154-f007:**
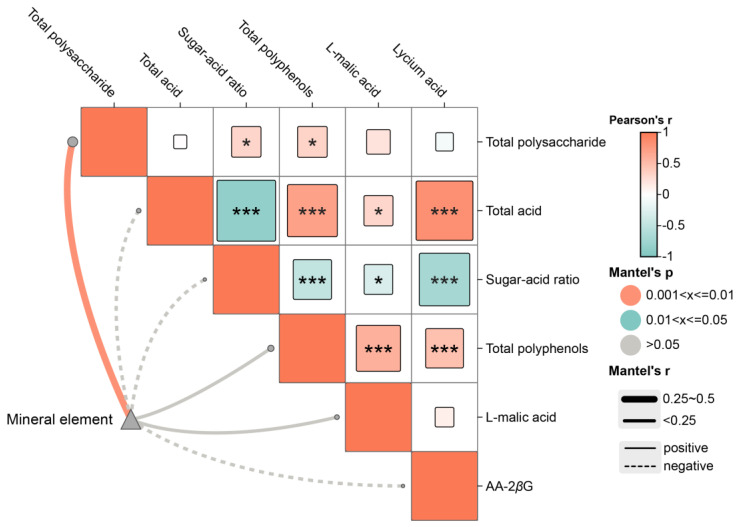
Mantel analysis of the quality indicators and medicinal properties of E’seguo. * Indicates *p* < 0.05, *** indicates *p* < 0.001.

**Figure 8 molecules-29-04154-f008:**
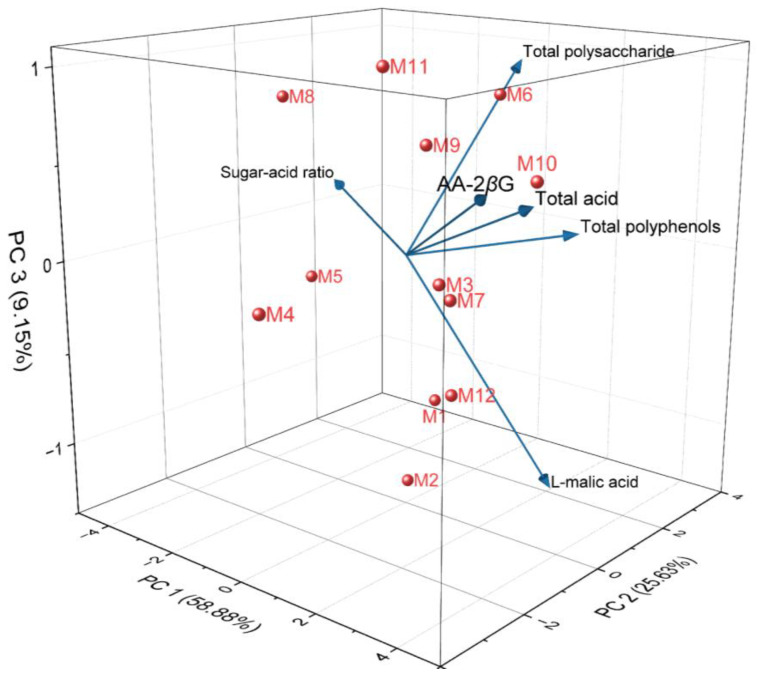
3D Principal Component Analysis (PCA) of the quality indicators and medicinal properties of E’seguo.

**Figure 9 molecules-29-04154-f009:**
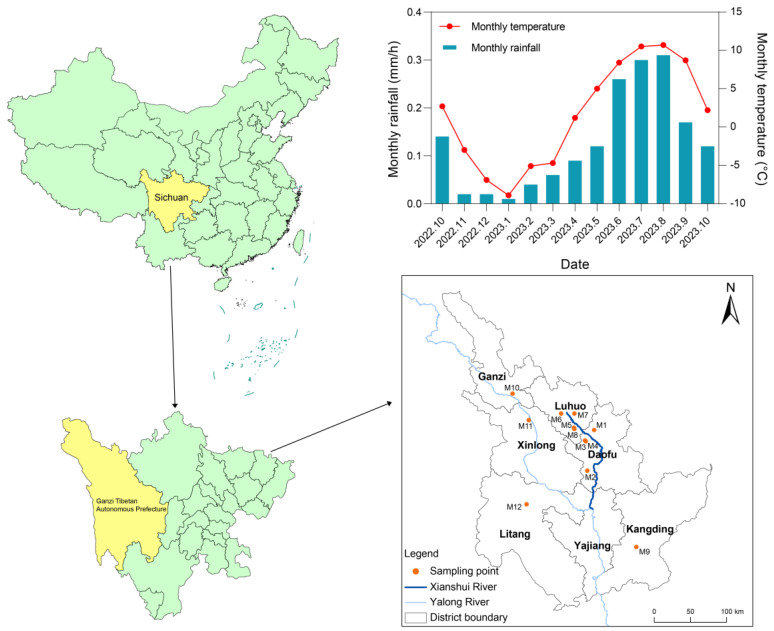
The specific sampling locations of E’seguo and the temperature and rainfall variations throughout the year.

**Figure 10 molecules-29-04154-f010:**
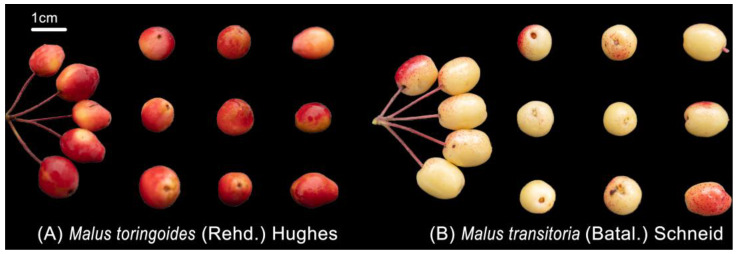
Harvest picture of the E’seguo. The length of the white line in the picture is 1 cm, (**A**) is *Malus toringoides* (Rehd.) Hughes, and (**B**) is *Malus transitoria* (Batal.) Schneid.

**Table 1 molecules-29-04154-t001:** Linear relationship between L-malic acid and AA-2*β*G.

Analyte	Equation	R^2^	Range (mg/mL)
L-malic acid	y = 17.734x − 3.4596	0.9990	0.626~1.565
AA-2*β*G	y = 395.97x + 2.2132	0.9991	0.031~0.760

**Table 2 molecules-29-04154-t002:** Principal Component Analysis (PCA) of the quality indicators and medicinal properties of E’seguo.

Traits		PC 1	PC 2	PC 3
Eigenvalue		3.533	1.538	0.549
Variability (%)		58.880	25.628	9.147
Cumulative (%)		58.880	84.508	93.655
Total polysaccharide	Factor loadings	0.041	0.906	0.386
Total acid		0.963	−0.124	0.172
Sugar-acid ratio		−0.888	0.333	0.097
Total polyphenols		0.865	0.389	0.070
L-malic acid		0.590	0.566	−0.560
AA-2*β*G		0.848	−0.345	0.206
Total polysaccharide	Component Score Coefficient Matrix (CSC)	0.022	0.730	0.521
Total acid		0.512	−0.100	0.233
Sugar-acid ratio		−0.472	0.269	0.131
Total polyphenols		0.460	0.314	0.094
L-malic acid		0.314	0.456	−0.756
AA-2*β*G		0.451	−0.279	0.278

**Table 3 molecules-29-04154-t003:** Score table of the Principal Component Analysis (PCA) for the quality indicators and medicinal properties of E’seguo.

Varieties	H_1_	H_2_	H_3_	H_0_
M1	−0.415	1.213	−0.936	−0.021
M2	−0.498	0.537	−1.358	−0.299
M3	1.276	−0.387	−0.077	0.689
M4	−0.572	−3.020	−0.134	−1.199
M5	−3.073	0.282	−0.273	−1.882
M6	0.362	2.166	0.774	0.896
M7	1.870	−0.698	−0.109	0.974
M8	−3.601	0.054	0.739	−2.177
M9	0.342	0.173	0.586	0.320
M10	2.922	0.415	0.472	1.997
M11	0.050	−0.642	1.008	−0.046
M12	1.336	−0.092	−0.692	0.747

**Table 4 molecules-29-04154-t004:** Information on the origin and varieties of E’seguo from different production areas. All samples were collected from Ganzi Prefecture, Sichuan Province, China.

Sample Number	Variety	Sampling Location	River Basin
M1	*Malus toringoides* (Rehd.) Hughes	Kongse Township, Daofu County	Xianshui river
M2	*Malus toringoides* (Rehd.) Hughes	Genji Village, Wari Town, Daofu County	Xianshui river
M3	*Malus toringoides* (Rehd.) Hughes	Mazi Township, Daofu County	Xianshui river
M4	*Malus transitoria* (Batal.) Schneid	Mazi Township, Daofu County	Xianshui river
M5	*Malus toringoides* (Rehd.) Hughes	Renda Township, Luhuo County	Xianshui river
M6	*Malus toringoides* (Rehd.) Hughes	Keke, Xiala Tuo Town, Luhuo County	Xianshui river
M7	*Malus toringoides* (Rehd.) Hughes	Yade Township, Luhuo County	Xianshui river
M8	*Malus transitoria* (Batal.) Schneid	Renda Township, Luhuo County	Xianshui river
M9	*Malus toringoides* (Rehd.) Hughes	Jiagenba Town, Kangding City	Yalong river
M10	*Malus toringoides* (Rehd.) Hughes	Tuoba Township, Ganzi County	Yalong river
M11	*Malus toringoides* (Rehd.) Hughes	Dagai Township, Xinlong County	Yalong river
M12	*Malus toringoides* (Rehd.) Hughes	Junba Town, Litang County	Yalong river

## Data Availability

The data presented in this study are available upon request from the corresponding author. The data are not publicly available due to privacy concerns.

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
