# Peer review of "Study on the Bioactive Constituent and Mineral Elements of the Tibetan Medicine E’seguo from Different Regions of Ganzi Prefecture, China"

_molecules, 2024, doi:10.3390/molecules29174154_

Round 1

Reviewer 1 Report

Comments and Suggestions for Authors

The paper of Jiang et al. “Study on the Bioactive Constituent and Mineral Elements …” aimed to provisional analysis of two Malus fruits. Generally, the article contains the results of a fairly simple analysis that do not lead to any general conclusions and reduced those small advantages that could be estimated as positive.

Highlights and strengths of the manuscript are:

The results may further increase interest in Malus toringoides and Malus transitoria fruits and help develop new strategies for medical application of E´seguo fruits.

Specific comments and suggested revisions:

Samples:

-          The elemental composition of plant tissues depends largely on climatic conditions, including water regime (precipitation, features of the mineral composition of the soil, etc.). To obtain reliable data on elemental composition, it is necessary to collect samples over several years, indicating information on weather and soil composition. The results you obtained cannot be extrapolated to samples from other regions due to the low repetition, which makes your results insignificant.

-          It is not clear what was “samples were de-enzymed” meant.

-          Why was such a large particle size of 4.75 mm chosen for analytical studies, which does not allow for complete extraction of target components?

-          Flora of China gives the following description of Malus fruits: for Malus transitoria (Batalin) C. K. Schneider – “Pome yellowish red” (http://www.efloras.org/florataxon.aspx?flora_id=2&taxon_id=200010919), and for Malus toringoides (Rehder) Hughes – “Pome yellow, tinged red” (http://www.efloras.org/florataxon.aspx?flora_id=2&taxon_id=200010918). Your figure 1 shows other colors. Please unify information about fruits description with botanical references. 

-          The different number of samples studied (10 samples for Malus toringoides and 2 for Malus transitoria) does not allow for a reliable comparison of the results.

-          What is the repetition for quantitative analysis?

Methods:

-          What is the reason for choosing these particular elements (K, Ca, Mg, Fe, Mn, Cu, Zn, Se) for the analysis of samples? In its current form, this set appears random.

-          Calculation of sugar-to-acid ratio for apples quality used mono+disaccharides content not polysaccharides content.

-          Can’t find any information about lycium acid. Description wkq20042702 refers to chlorogenic acid (https://xb.njucm.edu.cn/cn/article/pdf/preview/10.14148/j.issn.1672-0482.2024.0479.pdf). Is lycium acid is a chlorogenic acid? If so, please correct and use “usual” or systematical names for the common phytochemicals.

-          Description of malic acid extraction and analysis includes sentence “Other conditions were the same as for citric acid”. What did you mean?

Results:

As a result, the content of 8 random elements, total polysaccharides, total acids, total polyphenols, malic acid and an unknown acid were determined in 12 plant samples. After this, the data were statistically processed using various methods, which did not lead to any clear conclusion.

With all due respect to authors, I see no possibility to recommend paper for publication. The paper has serious flaws that can be corrected by adding new reliable data and acceptable discussion.

Author Response

Dear Reviewers,

We sincerely thank you for your thorough review and valuable suggestions on our manuscript. We have carefully considered all your comments and made the necessary revisions. The details of the modifications have been outlined in the attached Word document. Your feedback has been instrumental in improving the quality of our work, and we greatly appreciate your efforts.

Best regards,

Heling Fan

Reviewer 2 Report

Comments and Suggestions for Authors

2024_08_molecules-3161039-peer-review-v1 Medicine tibetane         

The manuscript deals with the analyses of two fruits, namely Malus toringoides (Rehd.) Hughes and Malus transitoria (Batal.) Schneid belonging to the Apple genus, Rosaceae family. These fruits are listed as Tibetan medicines with the name E’seguo. The fruits were collected in different regions in  Ganzi Prefecture, Sichuan Province, China.

The concentrations of mineral nutrients, polysaccharides, total acids, sugar-acid ratio, polyphenols and two acids (L-malic acid and Lycium acid, see below) were determined. The results were processed with chemometric techniques and the differences between the two fruits and between the regions in which they were collected were discussed. The authors concluded that Malus transitoria (Batal.) Schneid, owing to its composition, is sweeter and sourer and suitable for consumption, whereas Malus toringoides (Rehd.) Hughes is more suitable for medicinal use.

In my opinion the topic of the manuscript is of interest, owing to the increasing attention devoted to unconventional medicines all over the world. The extensive statistical processing of data gives Is useful to interpret the experimental results.

However, I think that the description of materials and methods is somewhat incomplete and confused, and the sections on Results should be written in a clearer way. For these reasons, I think the manuscript requires major revisions.

In particular:

- line 44 and elsewhere in the manuscript: “Lycium acid” is mentioned. I never heard of it. The formula and the systematic name of this acid should be reported

- Line 98: The authors wrote “The collected samples were de-enzymed and dried…” The procedure used to de-enzyme the samples should be briefly described

- line 109: the title of section 2.2 (Determination of polysaccharides, total acids, sugar-acid ratio, and polyphenols) does not correspond to the content of the section: the determination of mineral elements should also be mentioned in the title.

- line 128: the meaning of the abbreviations Wkq2102190 and Wkq20042702 should be explained

- sections 2.2 and 2.3: the procedure used for calibration (e.g. external calibration, internal standard or standard addition calibration) should be reported.

- section 2.3. The method for the determination of malic acid and Lycium acid is not properly described. The extraction medium is water, which is not selective at all, so it is possible that several peaks are present in the chromatograms: the retention times of the two analytes should be indicated. No validation of the method used for L-malic and Lycium acid determination is reported: at least a bibliographic reference on the method should be cited. Furthermore, in the procedure for lycium acid, citric acid is mentioned (line 131 and line 135)

- In general, no information on the reliability of the analytical methods (e.g precision or accuracy) described in section 2 is shown. At least, data taken from bibliographic references should be reported

- line 167: the term “indicator” in the title of section 3.1 and in the caption to Figure 5 is unclear. Its meaning should be explained in the text; otherwise, it might be deleted.

- lines 168-180: I suggest to collect the data on mineral elements in a Table, instead of listing them in the text of the manuscript. A discussion on the results might be focused on the comparison of element concentrations e.g. the ones with highest levels, the ones with lower levels and so on.

- Figures 2-10 ,Table 2 and Table 3 are not cited in the text. All of them should be mentioned.

- lines 197-198: for the sentence “Among the mineral elements, Mg, Ca, Se, and Zn were clustered together, while Fe, K, Mn, and Cu were grouped together”: the statistical approach used to identify the cluster should be reported. If this clustering is shown in a figure, the number of the figure should be cited.

- line 223: I am surprised to see that low values of r2, such as 0.29, indicate a significant correlation. Was the significance of the correlation found with a statistical software, or was it just hypothesized by the authors? The same considerations are valid for lines 275-282

- line 233: the meaning of “quality indicators” should be explained in the text of the manuscript

- lines 234-24e: I suggest to collect the data in a Table, instead of listing them in the text of the manuscript.

- line 281: a brief explanation on Mantel analysis should be given, since this technique is less common than PCA and HCA and many readers are probably non familiar with it.

- line 310: the procedure used for standardizing the quality data should be explained

- line 312: six principal components are mentioned here, but the reason why six (and not five or seven) were addressed is not clear.

- lines 346-347: the meaning of the abbreviations ICP-AES and AFS should be explained

- lines 346-370: I suppose the analysis of 565 apples refers to another study, different from the one reported in the present manuscript. If this is true, the bibliographic reference should be cited. I do not understand if all the sentences from line 346 to line 370 refer to such bibliographic reference, or if some of them refer to the study described in this manuscript. These aspects should be clarified.

- line  475: “MOLECULES” hould not be written in capital letters, but with the same style as the other journal names

Comments on the Quality of English Language

The English language is reasonable and only moderate revisions are required

Author Response

(The authors gave the same response as above.)

Round 2

Reviewer 1 Report

Comments and Suggestions for Authors

The authors taking into account the observations made by the reviewer. Considering the explanation provided by the authors the paper after correction may accepted in present from.

Author Response

(The authors gave the same response as above.)

Reviewer 2 Report

Comments and Suggestions for Authors

Comments on the revised manuscript molecules-3161039:  Study on the Bioactive Constituent and Mineral Elements of Tibetan Medicine E´seguo from Different Regions of Ganzi Prefecture, China

I think that the quality of the manuscript has much improved after the revisions made by the authors.

However, some of the responses do not mention the right line number, or refer to changes that were not made. Thus, in my opinion, the manuscript can be accepted after minor revisions.

In particular:

- Response 3: “The title has been supplemented with mineral element determination and is modified on line 140.”

Reviewer’s comment: the revised title (line 140) is: 2.2. Determination of mineral Indicator, polysaccharides, total acids, sugar-acid ratio, and poly-phenol

I think that the expression “mineral indicator” is not suitable in the context of the manuscript.

I searched for “Mineral indicator” on the Internet and found that “Indicator minerals are mineral species that, when appearing as transported grains in clastic sediments, indicate the presence in bedrock of a specific type of mineralization, hydrothermal alteration or lithology”

Instead of “mineral indicator”, the term “mineral elements” should be used in the title of chapter 2.2 (line 140), of chapter 3.1 (line 205), caption to Figure 6 (line 275), and in the Introduction (line 72)

- Response 6: “The retention times of L-malic acid and AA-2 β G were added, which are 8.1 and 9.2 minutes, respectively, and the modifications are shown in line 63-64. citric acid was a clerical error and has been revised to AA-2 β G, and the modifications are shown in lines 133 and 138”

Reviewer’s comment: The line numbers are wrong, but the modifications were made in the manuscript and appear in other lines

- Response 7. Reviewer’s comment: the text of response 7, i.e. the procedure and the investigation on repeatability, stability and recovery (from section 1 - Methodological investigation of total polysaccharides, to section 4.4 - Sample recovery rate investigation) and tables 1-18, is interesting and should be added as supplementary material, to avoid space problems. Of course, this supplementary material should be cited in the text of the manuscript.

- Response 8: "indicator" refers to the measured Mineral element indicators, namely K, Ga, Zn, Mg, Cu, Fe, Se, Mn8 elements, but because all 8 elements are written too much, so the text is expressed as "Mineral indicator".

Reviewer’s comment: Please see my comment to Response 3 and write “mineral elements” instead of “mineral element indicators”

- Response 9: The original data of each batch are shown in the table at the end of the article, but this paragraph is a comparative analysis of the data of different elements in each production area, so I think it should be retained

Reviewer’s comment: The Table is not reported at the end of the article, but at the end of the “response to reviewer 2 comments”.  (Table 19). Since it is quite long, it can be added as supplementary material, to avoid space problems. Of course, this table should be cited as supplementary material in the text of the manuscript.

- Response 10: “Figures 2-10 can not be found in the text, only Figure 2 and table 2, because some pictures and tables have been added, so their serial numbers become Figure 10 and table 4, see lines 337 and 366 for specific modifications”

Reviewer’s comment: no table or figure number is reported in lines 337 and 366. The table and figure numbering is right, but only Table 1 and 2 and Figures 1 and 2 are mentioned in the text of the manuscript. Also tables 3 and 4 and Figures 3-10 should be mentioned in the text of the manuscript.

- Response 11: “This sentence is derived from Figure 3 and has been modified in the text. See line 241.”

Reviewer’s comment: Figure 3 is not mentioned in line 241. Furthermore, the Figure to be cited in the sentence is Figure 4.

- Response 12: “The correlation data were derived from statistical software analysis and were not hypothesized by the authors, so the data are appropriate and valid”

Reviewer’s comment: I realize that the correlation data were derived from statistical software analysis and are appropriate and valid. My remark was about the significance of the correlations. Did the software indicate which correlations are significant?

- Response 18: The full name of ICP-AES andAFS is Inductively coupled plasma atomic emission spectrometry and Atomic fluorescence spectroscopy, which has been modified in the text, See lines 393-395

Reviewer’s comment: the meaning of the abbreviations has not been written in the text. It should be added in lines 393-395

- Response 19: “Only lines 346-350 belong to citations, which have been adjusted for the position of reference 29, see line 398”.

Reviewer’s comment: reference 29 is mentioned in line 405, which suggests that it refers to lines 393-403.

- Response 20: “Changes have been made, see line 476.”

Reviewer’s comment: line number 476 is wrong. The word MOLECULES in in line 535 and it is still in capital letter.

- Table 19: as I wrote in my comment to Response 9, Table 19 should be reported in the Supplementary material.

Author Response

(The authors gave the same response as above.)
